# AGGREGATED MOMENTUM:
# STABILITY THROUGH PASSIVE DAMPING

**James Lucas, Shengyang Sun, Richard Zemel, Roger Grosse**
University of Toronto; Vector Institute
{jlucas, ssy, zemel, rgrosse}@cs.toronto.edu

## ABSTRACT

Momentum is a simple and widely used trick which allows gradient-based optimizers to pick up speed along low curvature directions. Its performance depends crucially on a damping coefficient $\beta$. Large $\beta$ values can potentially deliver much larger speedups, but are prone to oscillations and instability; hence one typically resorts to small values such as 0.5 or 0.9. We propose *Aggregated Momentum (AggMo)*, a variant of momentum which combines multiple velocity vectors with different $\beta$ parameters. AggMo is trivial to implement, but significantly dampens oscillations, enabling it to remain stable even for aggressive $\beta$ values such as 0.999. We reinterpret Nesterov's accelerated gradient descent as a special case of AggMo and analyze rates of convergence for quadratic objectives. Empirically, we find that AggMo is a suitable drop-in replacement for other momentum methods, and frequently delivers faster convergence with little to no tuning.

## 1 INTRODUCTION

In spite of a wide range of modern optimization research, gradient descent with momentum and its variants remain the tool of choice in machine learning. Momentum methods can help the optimizer pick up speed along low curvature directions without becoming unstable in high-curvature directions. The simplest of these methods, classical momentum (Polyak, 1964), has an associated damping coefficient, $0 \leq \beta < 1$, which controls how quickly the momentum vector decays. The choice of $\beta$ imposes a tradeoff between speed and stability: in directions where the gradient is small but consistent, the terminal velocity is proportional to $1/(1 - \beta)$, suggesting that $\beta$ slightly less than 1 could deliver much improved optimization performance. However, large $\beta$ values are prone to oscillations and instability (O'Donoghue & Candes, 2015; Goh, 2017), requiring a smaller learning rate and hence slower convergence.

Finding a way to dampen the oscillations while preserving the high terminal velocity of large beta values could dramatically speed up optimization. Sutskever et al. (2013) found that Nesterov accelerated gradient descent (Nesterov, 1983), which they reinterpreted as a momentum method, was more stable than classical momentum for large $\beta$ values and gave substantial speedups for training neural networks. However, the reasons for the improved performance remain somewhat mysterious. O'Donoghue & Candes (2015) proposed to detect oscillations and eliminate them by resetting the velocity vector to zero. But in practice it is difficult to determine an appropriate restart condition.

In this work, we introduce *Aggregated Momentum (AggMo)*, a variant of classical momentum which maintains several velocity vectors with different $\beta$ parameters. AggMo averages the velocity vectors when updating the parameters. We find that this combines the advantages of both small and large $\beta$ values: the large values allow significant buildup of velocity along low curvature directions, while the small values dampen the oscillations, hence stabilizing the algorithm. AggMo is trivial to implement and incurs almost no computational overhead.

We draw inspiration from the physics literature when we refer to our method as a form of *passive damping*. Resonance occurs when a system is driven at specific frequencies but may be prevented through careful design (Goldstein, 2011). Passive damping can address this in structures by making use of different materials with unique resonant frequencies. This prevents any single frequency from producing catastrophic resonance. By combining several momentum velocities together we achieve a similar effect — no single frequency is driving the system and so oscillation is prevented.

In this paper we analyze rates of convergence on quadratic functions. We also provide theoretical convergence analysis showing that AggMo achieves converging average regret in online convex programming (Zinkevich, 2003). To evaluate AggMo empirically we compare against other commonly used optimizers on a range of deep learning architectures: deep autoencoders, convolutional networks, and long-term short-term memory (LSTM).

In all of these cases, we find that AggMo works as a drop-in replacement for classical momentum, in the sense that it works at least as well for a given $\beta$ parameter. But due to its stability at higher $\beta$ values, it often delivers substantially faster convergence than both classical and Nesterov momentum when its maximum $\beta$ value is tuned.

## 2 Background: momentum-based optimization

**Classical momentum**    We consider a function $f : \mathbb{R}^d \to \mathbb{R}$ to be minimized with respect to some variable $\boldsymbol{\theta}$. Classical momentum (CM) minimizes this function by taking some initial point $\boldsymbol{\theta}_0$ and running the following iterative scheme,

$$
\begin{aligned}
\mathbf{v}_t &= \beta \mathbf{v}_{t-1} - \nabla_\theta f(\boldsymbol{\theta}_{t-1}), \\
\boldsymbol{\theta}_t &= \boldsymbol{\theta}_{t-1} + \gamma_t \mathbf{v}_t,
\end{aligned}
\tag{1}
$$

where $\gamma_t$ denotes a learning rate schedule, $\beta$ is the damping coefficient and we set $\mathbf{v}_0 = 0$. Momentum can speed up convergence but it is often difficult to choose the right damping coefficient, $\beta$. Even with momentum, progress in a low curvature direction may be very slow. If the damping coefficient is increased to overcome this then high curvature directions may cause instability and oscillations.

**Nesterov momentum**    Nesterov's Accelerated Gradient (Nesterov, 1983; 2013) is a modified version of the gradient descent algorithm with improved convergence and stability. It can be written as a momentum-based method (Sutskever et al., 2013),

$$
\begin{aligned}
\mathbf{v}_t &= \beta \mathbf{v}_{t-1} - \nabla_\theta f(\boldsymbol{\theta}_{t-1} + \gamma_{t-1}\beta \mathbf{v}_{t-1}), \\
\boldsymbol{\theta}_t &= \boldsymbol{\theta}_{t-1} + \gamma_t \mathbf{v}_t.
\end{aligned}
\tag{2}
$$

Nesterov momentum seeks to solve stability issues by correcting the error made after moving in the direction of the velocity, $\mathbf{v}$. In fact, it can be shown that for a quadratic function Nesterov momentum adapts to the curvature by effectively rescaling the damping coefficients by the eigenvalues of the quadratic (Sutskever et al., 2013).

**Quadratic convergence**    We begin by studying convergence on quadratic functions, which have been an important test case for analyzing convergence behavior (Sutskever et al., 2013; O'Donoghue & Candes, 2015; Goh, 2017), and which can be considered a proxy for optimization behavior near a local minimum (O'Donoghue & Candes, 2015).

We analyze the behavior of these optimizers along the eigenvectors of a quadratic function in Figure 1. In the legend, $\lambda$ denotes the corresponding eigenvalue. In (a) we use a low damping coefficient ($\beta = 0.9$) while (b) shows a high damping coefficient ($\beta = 0.999$). When using a low damping coefficient it takes many iterations to find the optimal solution. On the other hand, increasing the damping coefficient from 0.9 to 0.999 causes oscillations which prevent convergence. When using CM in practice we seek the critical damping coefficient which allows us to rapidly approach the optimum without becoming unstable (Goh, 2017). On the other hand, Nesterov momentum with $\beta = 0.999$ is able to converge more quickly within high curvature regions than CM but retains oscillations for the quadratics exhibiting lower curvature.

## 3 Passive damping through Aggregated Momentum

**Aggregated Momentum**    We propose Aggregated Momentum (AggMo), a variant of gradient descent which aims to improve stability while providing the convergence benefits of larger damping coefficients. We modify the gradient descent algorithm by including several velocity vectors each with their own damping coefficient. At each optimization step these velocities are updated and then averaged to produce the final velocity used to update the parameters. This updated iterative procedure can be written as follows,

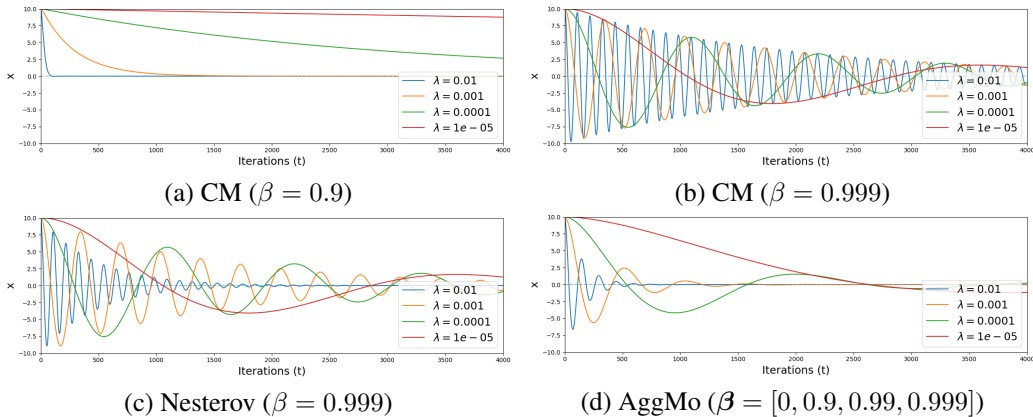

(a) CM ($\beta = 0.9$)        (b) CM ($\beta = 0.999$)

(c) Nesterov ($\beta = 0.999$)        (d) AggMo ($\boldsymbol{\beta} = [0, 0.9, 0.99, 0.999]$)

Figure 1: **Minimizing a quadratic function**. All optimizers use a fixed learning rate of 0.33. In the legend, $\lambda$ denotes the corresponding eigenvalues.

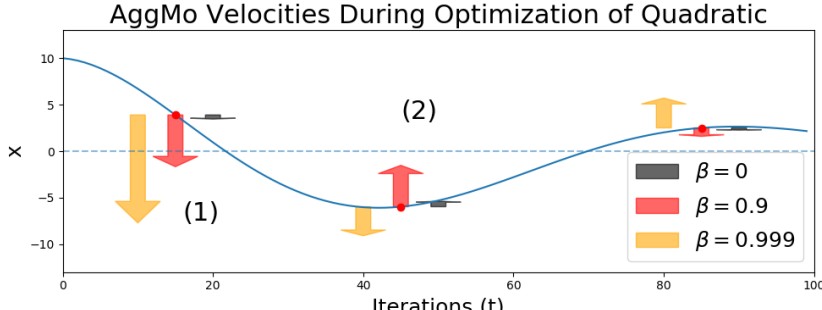

Figure 2: **Breaking oscillations with passive damping.** The arrows show the direction and relative amplitude of the velocities at various points in time. We discuss points (1) and (2) in Section 3.

$$\mathbf{v}_t^{(i)} = \beta^{(i)} \mathbf{v}_{t-1}^{(i)} - \nabla_\theta f(\boldsymbol{\theta}_{t-1}), \text{ for all } i,$$

$$\boldsymbol{\theta}_t = \boldsymbol{\theta}_{t-1} + \frac{\gamma_t}{K} \sum_{i=1}^{K} \mathbf{v}_t^{(i)}, \tag{3}$$

where $\mathbf{v}_0^{(i)} = 0$ for each $i$. We refer to the vector $\boldsymbol{\beta} = [\beta^{(1)}, \dots, \beta^{(K)}]$ as the *damping vector*.

By taking advantage of several damping coefficients, AggMo is able to optimize well over ill-conditioned curvature. Figure 1 (d) shows the optimization along the eigenvectors of a quadratic function using AggMo. AggMo dampens oscillations quickly for all eigenvalues and converges faster than CM and Nesterov in this case.

In Figure 2 we display the AggMo velocities during optimization. At point (1) the velocities are aligned towards the minima, with the $\beta = 0.999$ velocity contributing substantially more to each update. By point (2) the system has begun to oscillate. While the $\beta = 0.999$ velocity is still pointed away from the minima, the $\beta = 0.9$ velocity has changed direction and is damping the system. Combining the velocities allows AggMo to achieve fast convergence while reducing the impact of oscillations caused by large $\beta$ values.

## 3.1 Using AggMo

**Choosing the damping vector**    Recall that in a direction with small but steady gradient, the terminal velocity is proportional to $1/(1 - \beta)$. We found that a good choice of damping vectors was therefore to space the terminal velocities exponentially. To do so, we specify an exponential scale-factor, $a$, and a count $K$. The damping vector is then constructed as $\beta^{(i)} = 1 - a^{i-1}$, for $i = 1 \dots K$. We

fix $a = 0.1$ throughout and vary only $K$. A good default choice is $K = 3$ which corresponds to $\boldsymbol{\beta} = [0, 0.9, 0.99]$. We found this setting to be both stable and effective in all of our experiments.

**Computational/Memory overhead** There is very little additional computational overhead when using AggMo compared to CM, as it only requires a handful of extra addition and multipliciation operations on top of the single gradient evaluation. There is some memory overhead due to storing the $K$ velocity vectors, which are each the same size as the parameter vector. However, for most modern deep learning applications, the memory cost at training time is dominated by the activations rather than the parameters (Gomez et al., 2017; Chen et al., 2016; Werbos, 1990; Hochreiter & Schmidhuber, 1997), so the overhead will generally be small.

# 4 Recovering Nesterov momentum

In this section we show that we can recover Nesterov Momentum (Equation 2) using a simple generalization of Aggregated Momentum (Equation 3). We now introduce separate learning rates for each velocity, $\gamma^{(i)}$, so that the iterate update step from Equation 3 is replaced with,

$$\boldsymbol{\theta}_t = \boldsymbol{\theta}_{t-1} + \frac{1}{K} \sum_{i=1}^{K} \gamma_t^{(i)} \mathbf{v}_t^{(i)}, \tag{4}$$

with each velocity updated as in Equation 3. To recover Nesterov momentum we consider the special case of $\boldsymbol{\beta} = [0, \beta]$ and $\gamma_t^{(1)} = 2\gamma$, $\gamma_t^{(2)} = 2\beta\gamma$. The AggMo update rule can now be written as,

$$\begin{aligned}
\mathbf{v}_t &= \beta \mathbf{v}_{t-1} - \nabla_\theta f(\boldsymbol{\theta}_{t-1}), \\
\boldsymbol{\theta}_t &= \boldsymbol{\theta}_{t-1} + \frac{\gamma^{(2)}}{2} \mathbf{v}_t - \frac{\gamma^{(1)}}{2} \nabla_\theta f(\boldsymbol{\theta}_{t-1}), \\
&= \boldsymbol{\theta}_{t-1} + \gamma \beta^2 \mathbf{v}_{t-1} - (1 + \beta) \gamma \nabla_\theta f(\boldsymbol{\theta}_{t-1}).
\end{aligned} \tag{5}$$

Similarly, we may write the Nesterov momentum update with constant learning rate $\gamma_t = \gamma$ as,

$$\begin{aligned}
\mathbf{v}_t &= \beta \mathbf{v}_{t-1} - \nabla_\theta f(\boldsymbol{\theta}_{t-1} + \gamma \beta \mathbf{v}_{t-1}), \\
\boldsymbol{\theta}_t &= \boldsymbol{\theta}_{t-1} + \gamma \beta \mathbf{v}_{t-1} - \gamma \nabla_\theta f(\boldsymbol{\theta}_{t-1} + \gamma \beta \mathbf{v}_{t-1}).
\end{aligned} \tag{6}$$

Now we consider Equation 6 when using the reparameterization given by $\boldsymbol{\phi}_t = \boldsymbol{\theta}_t + \gamma \beta \mathbf{v}_t$,

$$\begin{aligned}
\boldsymbol{\phi}_t - \gamma \beta \mathbf{v}_t &= \boldsymbol{\phi}_{t-1} - \gamma \nabla_\theta f(\boldsymbol{\phi}_{t-1}), \\
\Rightarrow \boldsymbol{\phi}_t &= \boldsymbol{\phi}_{t-1} + \gamma \beta \mathbf{v}_t - \gamma \nabla_\theta f(\boldsymbol{\phi}_{t-1}), \\
&= \boldsymbol{\phi}_{t-1} + \gamma \beta^2 \mathbf{v}_{t-1} - (1 + \beta) \gamma \nabla_\theta f(\boldsymbol{\phi}_{t-1}).
\end{aligned} \tag{7}$$

It follows that the update to $\phi$ from Nesterov is identical to the AggMo update to $\boldsymbol{\theta}$, and we have $\boldsymbol{\phi}_0 = \boldsymbol{\theta}_0$. We can think of the $\phi$ reparameterization as taking a half-step forward in the Nesterov optimization allowing us to directly compare the iterates at each time step. We note also that if $\gamma_t^{(1)} = \gamma_t^{(2)} = 2\gamma$ then the equivalence holds approximately when $\beta$ is sufficiently close to 1. We demonstrate this equivalence empirically in Appendix B.

This formulation allows us to reinterpret Nesterov momentum as a weighted average of a gradient update and a momentum update. Moreover, by showing that AggMo recovers Nesterov momentum we gain access to the same theoretical convergence results that Nesterov momentum achieves.

# 5 Convergence analysis

## 5.1 Analyzing quadratic convergence

We can learn a great deal about optimizers by carefully reasoning about their convergence on quadratic functions. O'Donoghue & Candes (2015) point out that in practice we do not know the condition number of the function to be optimized and so we aim to design algorithms which work well over a

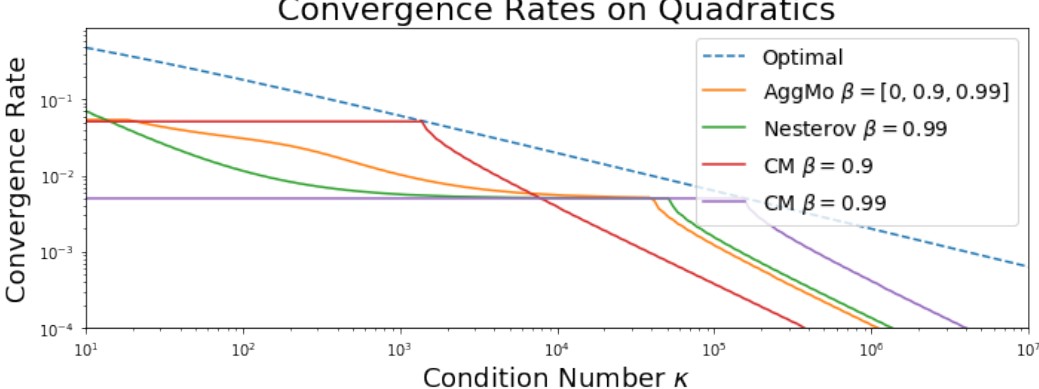

Figure 3: **Convergence on quadratics of varying condition number.** AggMo interpolates between the convergence rates of CM at $\beta = 0.9$ and $\beta = 0.99$.

large possible range. Sharing this motivation, we consider the convergence behaviour of momentum optimizers on quadratic functions with fixed hyperparameters over a range of condition numbers.

To compute the convergence rate, $||\boldsymbol{\theta}_t - \boldsymbol{\theta}^*||^2$, we model each optimizer as a linear dynamical systems as in Lessard et al. (2016). The convergence rate is then determined by the eigenvalues of this system. We leave details of this computation to appendix B.

Figure 3 displays the convergence rate of each optimizer for quadratics with condition numbers ($\kappa$) from $10^1$ to $10^7$. The blue dashed line displays the optimal convergence rate achievable by CM with knowledge of the condition number — an unrealistic scenario in practice. The two curves corresponding to CM (red and purple) each meet the optimal convergence rate when the condition number is such that $\beta$ is critical. On the left of this critical point, where the convergence rates for CM are flat, the system is "under-damped" meaning there are complex eigenvalues corresponding to oscillations.

We observe that the convergence rate of AggMo interpolates smoothly between the convergence rates of CM with $\beta = 0.9$ and $\beta = 0.99$ as the condition number varies. AggMo's ability to quickly kill oscillations leads to an approximately three-times faster convergence rate than Nesterov momentum in the under-damped regime without sacrificing performance on larger condition numbers.

## 5.2 Additional convergence analysis

We evaluate the convergence rate of AggMo in the setting of online convex programming, as proposed in Zinkevich (2003). This is an increasingly common setting to analyze optimization algorithms tailored to machine learning (Duchi et al., 2011; Kingma & Ba, 2014; Reddi et al., 2018). Notably, this is equivalent to analyzing the convergence rate in the setting of stochastic convex optimization.

We consider a sequence of unknown convex cost functions, $f_1(\boldsymbol{\theta}), \ldots, f_T(\boldsymbol{\theta})$. At each time $t$, our goal is to predict the parameter $\boldsymbol{\theta}_t$ which minimizes the regret,

$$R(T) = \sum_{t=1}^{T} \left[ f_t(\theta_t) - f_t(\theta^*) \right], \tag{8}$$

where $\boldsymbol{\theta}^*$ is the fixed point parameter minimizing $\sum_{t=1}^{T} f_t(\theta*)$. We are able to show that AggMo has regret bounded by $O(\sqrt{T})$ - a result asymptotically comparable to the best known bound (Duchi et al., 2011). We adopt the following definitions from Duchi et al. (2011) to simplify the notation. We write $g_t = \nabla f_t(\boldsymbol{\theta}_t)$ with $g_{t,i}$ as the $i^{th}$ element of this vector. Additionally, we write $g_{1:t,i} \in \mathbb{R}^t$ as the vector containing the $i^{th}$ element of the gradient over the first $t$ iterations; $g_{1:t,i} = [g_{1,i}, \ldots, g_{t,i}]$. Then the following theorem holds,

**Theorem 1.** *Assume that $f_t$ has bounded gradients, $||\nabla f_t(\boldsymbol{\theta})||_2 < G, ||\nabla f_t(\boldsymbol{\theta})||_\infty < G_\infty, \forall \boldsymbol{\theta} \in \mathbb{R}^d$. Moreover, assume that each $\boldsymbol{\theta}_t$ generated by AggMo satisfies $||\boldsymbol{\theta}_n - \boldsymbol{\theta}_m||_2 \leq D, ||\boldsymbol{\theta}_n - \boldsymbol{\theta}_m||_\infty \leq D_\infty$ for all $m, n \in \{1, \ldots, T\}$. Let $\gamma_t = \dfrac{\gamma}{\sqrt{t}}$ and $\beta_t^{(i)} = \beta^{(i)} \lambda^t, \lambda \in (0, 1)$. Then AggMo achieves the following regret bound, for all $T \geq 1$.*

$$R(T) \leq \frac{D_\infty^2 \sqrt{T}}{\gamma} + \frac{\gamma \sqrt{1 + \log(T)}}{2K} \sum_{j=1}^d ||g_{1:T,j}||_4^2 \sum_{i=1}^K \frac{1 + \beta^{(i)}}{(1 - \beta^{(i)})^2} + \frac{D^2}{2K\gamma(1 - \lambda)^2} \sum_{i=1}^K \beta^{(i)}.$$

It immediately follows that the average regret of AggMo converges, i.e. that $R(T)/T \to 0$, by observing that $||g_{1:T,j}||_4^2 \leq G_\infty^2 \sqrt{T}, \forall j$. The full proof is given in Appendix C alongside some open questions on the convergence of AggMo.

While the statement of Theorem 1 requires strict assumptions we note that this result is certainly non-trivial. Reddi et al. (2018) showed that the average regret of Adam (Kingma & Ba, 2014) is not guaranteed to converge under the same assumptions.

# 6   Related work

The convergence of momentum methods has been studied extensively, both theoretically and empirically (Wibisono & Wilson, 2015; Wibisono et al., 2016; Wilson et al., 2016; Kidambi et al., 2018). By analyzing the failure modes of existing methods these works motivate successful momentum schemes. Sutskever et al. (2013) explored the effect of momentum on the optimization of neural networks and introduced the momentum view of Nesterov's accelerated gradient. They focused on producing good momentum schedules during optimization to adapt to ill-conditioned curvature. Despite strong evidence that this approach works well, practitioners today still typically opt for a fixed momentum schedule and vary the learning rate instead.

In Appendix C.1 we show that AggMo evolves as a (K+1)-th order finite difference equation, enabling AggMo to utilize greater expressiveness over the gradient history. Liang et al. (2016) also introduce dependence on a larger gradient history by adding lagged momentum terms. However, in doing so the authors introduce many new hyperparameters to be tuned.

Adaptive gradient methods have been introduced to deal with the ill-conditioned curvature that we often observe in deep learning (Duchi et al., 2011; Kingma & Ba, 2014; Zeiler, 2012; Tieleman & Hinton, 2012). These methods typically approximate the local curvature of the objective to adapt to the geometry of the data. Natural gradient descent (Amari, 1998) preconditions by the Fisher information matrix, which can be shown to approximate the Hessian under certain assumptions (Martens, 2014). Several methods have been proposed to reduce the computational and memory cost of this approach (Martens & Grosse, 2015; Martens, 2010) but these are difficult to implement and introduce additional hyperparameters and computational overhead compared to SGD.

Another line of adaptive methods seeks to detect when oscillations occur during optimization. O'Donoghue & Candes (2015) proposed using an adaptive restarting scheme to remove oscillations whenever they are detected. In its simplest form, this is achieved by setting the momentum velocity to zero whenever the loss increases. Further work has suggested using an adaptive momentum schedule instead of zeroing (Srinivasan et al., 2018). Although this technique works well for well-conditioned convex problems it is difficult to find an appropriate restart condition for stochastic optimization where we do not have an accurate computation of the loss. On the other hand, AggMo's passive damping approach addresses the oscillation problem without the need to detect its occurrence.

# 7   Evaluation

We evaluated the AggMo optimizer on the following deep learning architectures; deep autoencoders, convolutional networks, and LSTMs. To do so we used four datasets: MNIST (LeCun et al., 1998), CIFAR-10, CIFAR-100 (Krizhevsky & Hinton, 2009) and Penn Treebank (Marcus et al., 1993). In each experiment we compared AggMo to classical momentum, Nesterov momentum, and Adam. These optimizers are by far the most commonly used and even today remain very difficult to outperform in a wide range of tasks. For each method, we performed a grid search over the learning rate and the damping coefficient. For AggMo, we keep the scale $a = 0.1$ fixed and vary $K$ as discussed in Section 3.1. Full details of the experimental set up for each task can be found in Appendix D with additional results given in Appendix E.

For each of the following experiments we choose to report the validation and test performance of the network in addition to the final training loss when it is meaningful to do so. We include these generalization results because recent work has shown that the choice of optimizer may have a significant effect on the generalization error of the network in practice (Wilson et al., 2017).

| Optimizer | Train Optimal | Validation Optimal | |
|---|---|---|---|
| | **Train Loss** | **Val. Loss** | **Test Loss** |
| **CM** | $2.51 \pm 0.06$ | $3.55 \pm 0.15$ | $3.45 \pm 0.15$ |
| **Nesterov** | $1.52 \pm 0.02$ | $3.20 \pm 0.01$ | $3.13 \pm 0.02$ |
| **Adam** | $1.44 \pm 0.02$ | $3.80 \pm 0.04$ | $3.72 \pm 0.05$ |
| **AggMo** | $\mathbf{1.39} \pm 0.02$ | $\mathbf{3.05} \pm 0.03$ | $\mathbf{2.96} \pm 0.03$ |

Table 1: **MNIST Autoencoder** We display the training MSE for the hyperparameter setting that achieved the best training loss. The validation and test errors are displayed for the hyperparameter setting that achieved the best validation MSE. In each case the average loss and standard deviation over 15 runs is displayed.

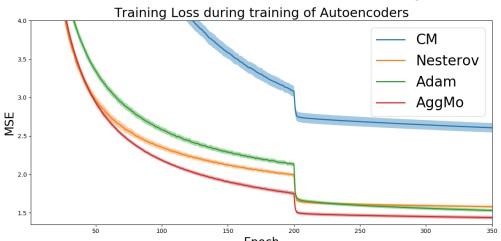
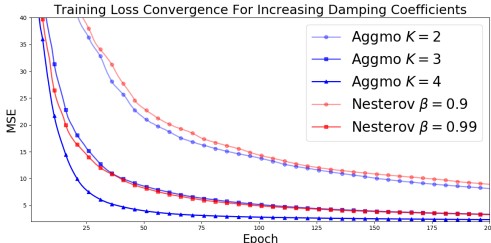

Figure 4: **Convergence of Autoencoders** Training loss during the first 350 epochs of training with each optimizer. The shaded region corresponds to one standard deviation over 15 runs.

Figure 5: **Damping Coefficient Investigation** Optimizing autoencoders on MNIST with varying damping coefficients and fixed learning rate. Nesterov is unstable with $\beta = 0.999$.

## 7.1 Autoencoders

We trained fully-connected autoencoders on the MNIST dataset using a set-up similar to that of Sutskever et al. (2013). While their work focused on finding an optimal momentum schedule we instead kept the momentum fixed and applied a simple learning rate decay schedule. For CM and Nesterov we evaluated damping coefficients in the range: $\{0.0, 0.9, 0.99, 0.999\}$. For Adam, it is standard to use $\beta_1 = 0.9$ and $\beta_2 = 0.999$. Since $\beta_1$ is analogous to the momentum damping parameter, we considered $\beta_1 \in \{0.9, 0.99, 0.999\}$ and kept $\beta_2 = 0.999$. For AggMo, we explored $K$ in $\{ 2,3,4 \}$. Each model was trained for 1000 epochs.

We report the training, validation, and test errors in Table 1. Results are displayed for the hyperparameters that achieved the best training loss and also for those that achieved the best validation loss. While Adam is able to perform well on the training objective it is unable to match the performance of AggMo or Nesterov on the validation/test sets. AggMo achieves the best performance in all cases.

In these experiments the optimal damping coefficient for both CM and Nesterov was $\beta = 0.99$ while the optimal damping vector for AggMo was $\boldsymbol{\beta} = [0.0, 0.9, 0.99, 0.999]$, given by $K = 4$. In Figure 4 we compare the convergence of each of the optimizers under the optimal hyperparameters for the training loss.

**Increasing damping coefficients** During our experiments we observed that AggMo remains stable during optimization for learning rates an order of magnitude (or more) larger than is possible for CM and Nesterov with $\beta$ equal to the max damping coefficient used in AggMo.

We further investigated the effect of increasing the maximum damping coefficient of AggMo in Figure 5. The learning rate is fixed at 0.1 and we vary $K$ from 2 to 5. We compared to Nesterov with damping coefficients in the same range (max of 0.9999) and a fixed learning rate of 0.05 (to be consistent with our analysis in Section 4). We do not include the curves for which training is unstable: Nesterov with $\beta \in \{0.999, 0.9999\}$ and AggMo with $K = 5$. AggMo is able to take advantage of the larger damping coefficient of 0.999 and achieves the fastest overall convergence.

## 7.2 Classification

For the following experiments we evaluated AggMo using two network architectures: a neural network with 5 convolutional layers (CNN-5) and the ResNet-32 architecture (He et al., 2016). We use data augmentation and regularization only for the latter. Each model was trained for 400 epochs.

| Optimizer | CNN-5 (CIFAR-10) | | ResNet-32 (CIFAR-10) | | ResNet-32 (CIFAR-100) | |
|---|---|---|---|---|---|---|
| | Val. (%) | Test (%) | Val. (%) | Test (%) | Val. (%) | Test (%) |
| CM | 64.1 | 63.43 | **94.20** | 93.16 | **70.38** | **70.21** |
| Nesterov | 65.14 | 64.32 | 94.16 | 93.18 | 70.34 | 70.08 |
| Adam | 63.67 | 62.86 | 92.36 | 90.94 | 67.20 | 68.08 |
| AggMo | **65.98** | **65.09** | 93.87 | 93.16 | 70.28 | 70.11 |
| CM ($\beta = 0.9$) | 64.1 | 63.43 | 94.10 | **93.36** | **70.38** | **70.21** |
| Nesterov ($\beta = 0.9$) | 64.13 | 63.04 | 94.16 | 93.18 | 70.34 | 70.08 |
| AggMo (Default) | **65.98** | **65.09** | 93.87 | 93.16 | 70.28 | 70.11 |

Table 2: **Classification accuracy on CIFAR-10 and CIFAR-100** We display results using the optimal hyperparameters for CM, Nesterov, Adam and AggMo on the validation set and also with default settings for CM, Nesterov and AggMo.

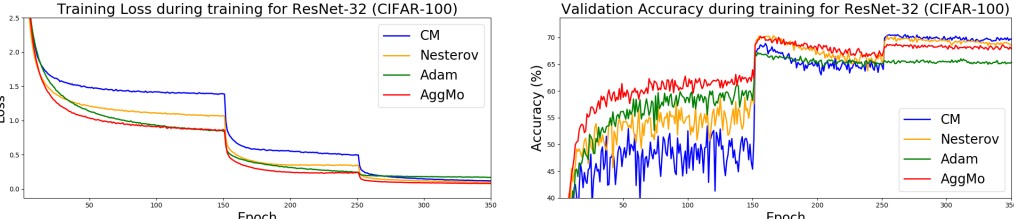

Figure 6: **ResNet-32 Trained On CIFAR-100** The training loss and validation accuracy during training on CIFAR-100 for each optimizer.

For each optimizer we report the accuracy on a randomly held out validation set and the test set. All of the models achieve near-perfect accuracy on the training set and so we do not report this. The results are displayed in Table 2. On the small convolutional network without regularization, AggMo significantly out performed the other methods. For both of the ResNet-32 experiments we observed the best validation accuracy with CM. This is perhaps expected as the model architecture and hyperparameters were likely to have been tuned using CM. Despite this, we observed that AggMo performed consistently well and had the fastest overall convergence.

We found that our proposed default hyperparameters for AggMo ($a = 0.1$, $K = 3$) led to much faster convergence than CM and Nesterov with $\beta = 0.9$, a common default choice. Figure 6 shows the training loss and validation accuracy during training for each optimizer used to train the ResNet-32 model. The hyperparameters used for each plot are those which obtained the best validation accuracy. AggMo converged most quickly on the training objective without sacrificing final validation performance.

Surprisingly, we found that using AggMo we were also able to train the ResNet-32 architecture on CIFAR-100 without using batch normalization. With a limited search over learning rates we achieved 69.32% test error compared to a best value of 67.26% using CM. We also found that, with batch normalization removed, optimization with AggMo remained stable at larger learning rates than with CM.

We note that the additional network hyperparameters (e.g. weight decay) are defaults which were likely picked as they work well with classical momentum. This may disadvantage the other optimizers, including our own. Despite this, we found that we are able to outperform CM with the AggMo and Nesterov optimizers without additional tuning of any of these hyperparameters.

## 7.3 Language modeling

We trained LSTM Language Models on the Penn Treebank dataset. We followed the experimental setup of Merity et al. (2017) and made use of the code provided by the authors. We used the optimal hyperparameter settings described by the authors and vary only the learning rate, momentum and whether gradient clipping is used. The network hyperparameters were tuned using SGD and may not be optimal for the other optimizers we evaluate (including our own). We followed only the base model training used in Merity et al. (2017) and do not include the fine-tuning and continuous cache optimization steps. Each model was trained for 750 epochs.

As noted in Merity et al. (2017), it is typically observed that SGD without momentum performs better than momentum-based methods in language modeling tasks. However, in our experiments we

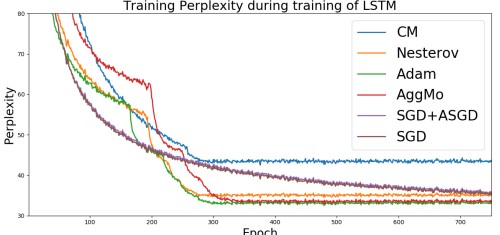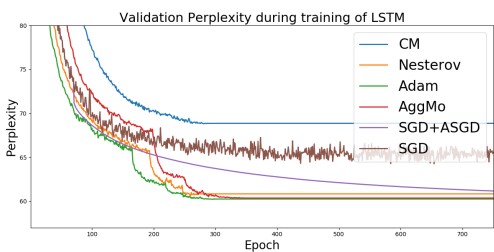

Figure 7: **Convergence of LSTM** The training and validation perplexity during training. For each model we use the hyperparameters that obtained the best validation loss. We found that there was very little difference when choosing hyperparameters based on training performance.

| Optimizer | Train Perplexity | Val. Perplexity | Test Perplexity |
|---|---|---|---|
| **\*SGD + ASGD** | 35.68 | 61.17 | 59.26 |
| **SGD** | 35.34 | 63.39 | 62.41 |
| **CM** | 50.34 | 70.37 | 68.21 |
| **Nesterov** | 34.91 | 60.84 | 58.44 |
| **Adam** | **32.88** | **60.25** | 57.83 |
| **AggMo** | 33.22 | 60.36 | **57.79** |

Table 3: **Penn Treebank LSTM** Perplexity across different optimizers. We display the train, validation, and test error for the optimization run that produced the best validation loss. * uses ASGD (Polyak & Juditsky, 1992) and corresponds to the base model reported in Merity et al. (2017)

observed all momentum-based optimizers but CM outperform SGD without momentum. Surprisingly, we found that Adam is well-suited to this task and achieves the best training, validation, and test performance. We believe that the heavy regularization used when training the network makes Adam a good choice. AggMo is very close in terms of final performance to Adam.

Table 3 contains the results for the hyperparameter settings which achieved the best validation error for each optimizer. The first row (denoted *) uses the scheme suggested in Merity et al. (2017): once the validation loss plateaus we switch to the ASGD (Polyak & Juditsky, 1992) optimizer. The other rows instead decay the learning rate when the validation loss plateaus.

Figure 7 compares the convergence of the training and validation perplexity of each optimizer. While the momentum methods converge after 300 epochs, the momentum-free methods converged much more slowly. Surprisingly, we found that SGD worked best without any learning rate decay. Adam converged most quickly and achieved a validation perplexity which is comparable to that of AggMo. While gradient clipping is critical for SGD without momentum, which utilizes a large learning rate, we found that all of the momentum methods perform better without gradient clipping.

In short, while existing work encourages practitioners to avoid classical momentum we found that using other momentum methods may significantly improve convergence rates and final performance. AggMo worked especially well on this task over a large range of damping coefficients and learning rates.

# 8    Conclusion

Aggregated Momentum is a simple extension to classical momentum which is easy to implement and has negligible computational overhead on modern deep learning tasks. We showed empirically that AggMo is able to remain stable even with large damping coefficients and enjoys faster convergence rates as a consequence of this. Nesterov momentum can be viewed as a special case of AggMo. (Incidentally, we found that despite its lack of adoption by deep learning practitioners, Nesterov momentum also showed substantial advantages compared to classical momentum.) On the tasks we explored, AggMo could be used as a drop-in replacement for existing optimizers with little-to-no additional hyperparameter tuning. But due to its stability at higher $\beta$ values, it often delivered substantially faster convergence than both classical and Nesterov momentum.

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

# Appendices

## A   Nesterov Equivalence

In this section we demonstrate this equivalence on two toy problems. In each of the figures included here we take $\boldsymbol{\beta} = 0.999$.

We first consider a 2D quadratic function, $f(\mathbf{x}) = \mathbf{x}^T A \mathbf{x}$, where $A$ has eigenvalues 1.0 and 0.001.The learning rates for each optimizer are set as described in Section 4. Each optimizer is initialized at the same position. Figure 8 shows both optimizers following the same optimization trajectories. In this setting, the two paths are also visually indistinguishable with $\gamma_t^{(1)} = \gamma_t^{(2)} = 2\gamma$ for AggMo.

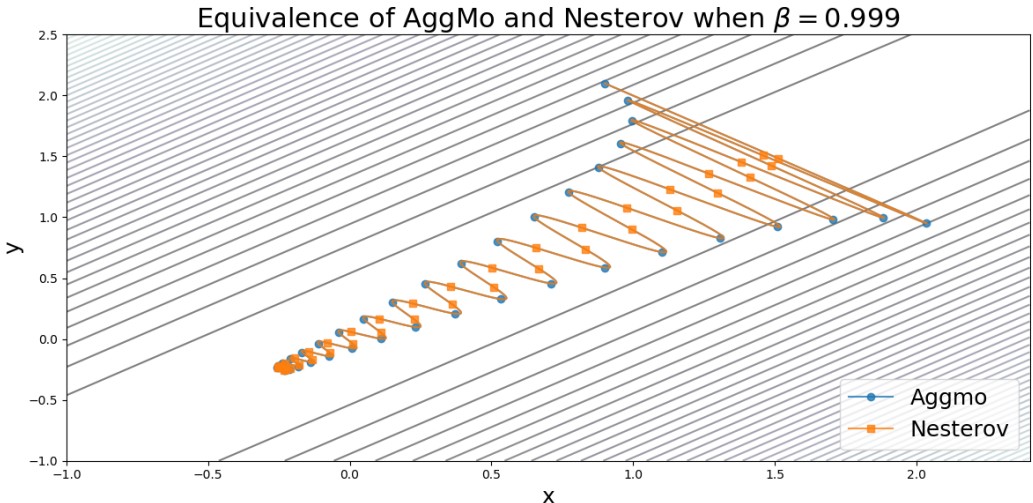

Figure 8: Equivalence of Nesterov and AggMo when $\beta = 0.999$. The optimization plots for $f(x) = \mathbf{x}^T A \mathbf{x}$ are visibly identical (circles correspond to AggMo and squares to Nesterov - the markers are offset for readability).

We now optimize the Rosenbrock function, given by,

$$f(x, y) = (y - x^2)^2 + 100(x - 1)^2$$

This function has a global minimum at $(x, y) = 1$. Once again the optimizers are initialized at the same point but for this example we take $\gamma_t^{(1)} = \gamma_t^{(2)} = 2\gamma$ for AggMo. Figure 9 shows the optimization trajectories of both algorithms. In this case we see that the updates are initially indistinguishable but begin to differ as the algorithms approach the origin.

## B   Quadratic Convergence Analysis

In this section we present details of the convergence rate computations in Figure 3. We also present some additional supporting results.

We first note that for quadratic functions of the form $f(\mathbf{x}) = \frac{1}{2}\mathbf{x}^T A x + b^T \mathbf{x}$ we can write the AggMo optimization procedure as a linear dynamical systems in $K + 1$ variables:

$$\begin{bmatrix} \mathbf{v}_{t+1}^{(1)} \\ \vdots \\ \mathbf{v}_{t+1}^{(K)} \\ \mathbf{x}_{t+1} - \mathbf{x}^* \end{bmatrix} = B \begin{bmatrix} \mathbf{v}_t^{(1)} \\ \vdots \\ \mathbf{v}_t^{(K)} \\ \mathbf{x}_t - \mathbf{x}^* \end{bmatrix}$$

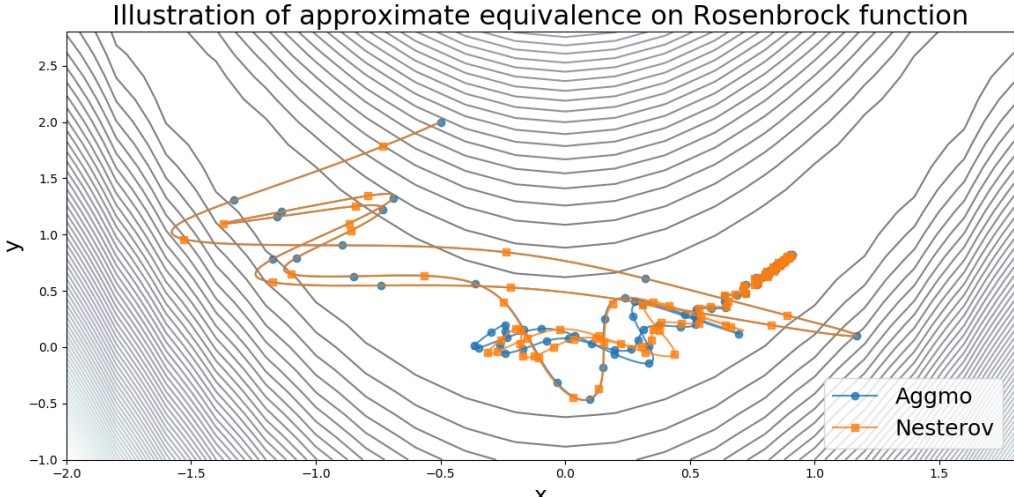

Figure 9: Approximate equivalence of Nesterov and AggMo when $\beta = 0.999$. The optimization trajectories are initially visibly identical but begin to differ slightly after more iterations.

The spectral norm of the matrix $B$ determines the rate at which the linear dynamical system converges and thus bounds $||\mathbf{x}_t - \mathbf{x}^*||^2$ (Lessard et al., 2016). We can write down the exact form of $B$ as follows,

$$B = \begin{bmatrix} \beta^{(1)} I & 0 & \cdots & 0 & -A \\ 0 & \beta^{(2)} I & \ddots & \vdots & \vdots \\ \vdots & \ddots & \ddots & 0 & -A \\ 0 & \cdots & 0 & \beta^{(K)} I & -A \\ \frac{\gamma\beta^{(1)}}{K} I & \frac{\gamma\beta^{(2)}}{K} I & \cdots & \frac{\gamma\beta^{(K)}}{K} I & (I - \gamma A) \end{bmatrix}$$

We note in particular that in the special case of $K = 1$ (CM) we recover the characteristic equation of O'Donoghue & Candes (2015):

$$u^2 - (1 + \beta - \gamma\lambda_i)u + \beta = 0$$

Which in turn yields the critical damping coefficient and optimal rate, with

$$\beta^* = \left( \frac{\sqrt{\kappa} - 1}{\sqrt{\kappa} + 1} \right)^2 .$$

When $\beta < \beta^*$ the system is over-damped and exhibits slow monotone convergence (Figure 1 (a)). When $\beta > \beta^*$ the system is under-damped and the characteristic equation yields imaginary solutions that correspond to oscillations (Figure 1 (b)) with convergence rate equal to $1 - |\beta|$. At the critical damping coefficient the convergence is optimal at $1.0 - \frac{\sqrt{\kappa} - 1}{\sqrt{\kappa} + 1}$.

We can combine this analysis with Theorem 2 from Sutskever et al. (2013) to recover similar convergence bounds for Nesterov momentum.

**Producing Figure 3**  To produce the curves in Figure 3 we compute the eigenvalues directly from the matrix $B$ for matrices $A$ with varying condition numbers. While we can find the optimal learning rate for CM and Nesterov momentum in closed form we have been unable to do so for AggMo. Therefore, we instead perform a fine-grained grid search to approximate the optimal learning rate for each condition number.

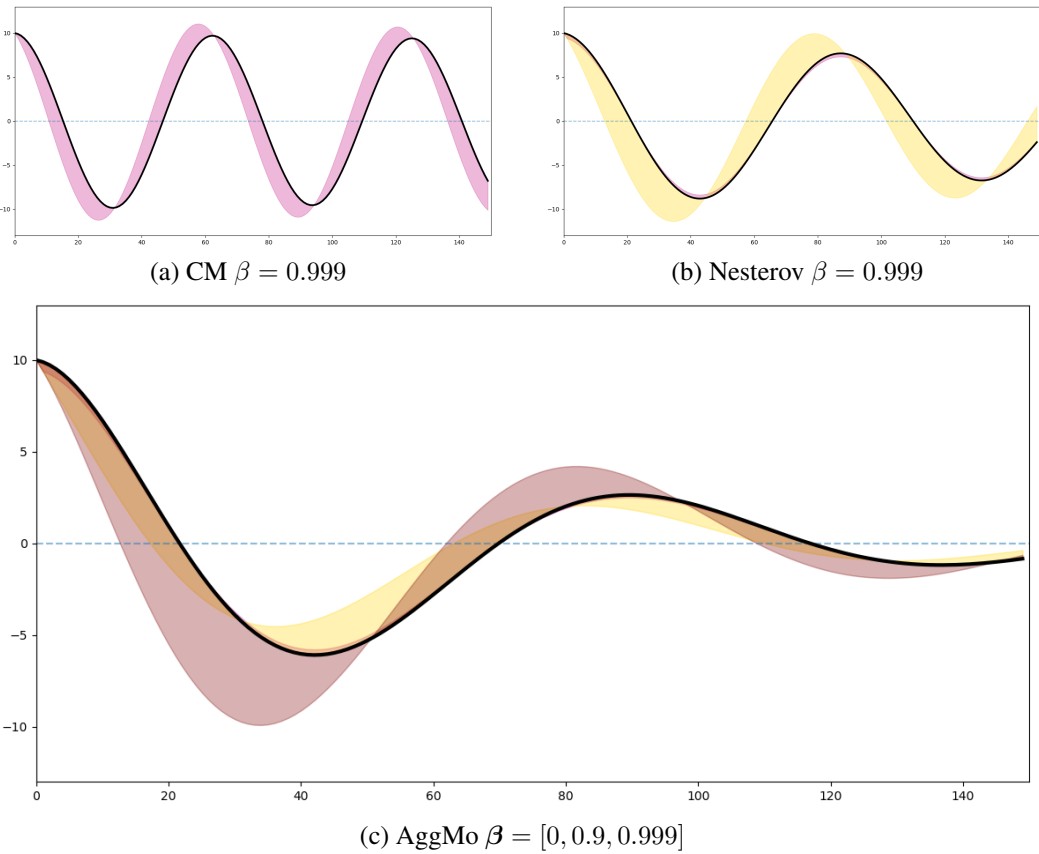

Figure 10: **Velocity during quadratic optimization with CM, Nesterov, and AggMo.** (Best viewed in color) The shaded region shows the direction and relative magnitude of the velocities throughout optimization for each optimizer. AggMo has multiple shaded regions corresponding to the different velocities.

**Studying Velocity**    We now present a brief study illustrating how using multiple velocities can break oscillations during optimization.

Figure 10 shows the optimization of a 1-D quadratic function with CM, Nesterov, and AggMo. The shaded region around each curve represents the direction and relative magnitude of the velocities term during optimization. CM (a) has a single velocity and oscillates at a near-constant amplitude. For Nesterov momentum (b) we display the velocity and the "error-correcting" term. AggMo (c) has shaded regions for each velocity. For AggMo, the velocity with $\beta = 0.9$ oscillates at a higher frequency and thus damps the whole system.

## C    Convergence Proof

Here we present the proof of Theorem 5 1. We introduce some simplifying notation used in Duchi et al. (2011). We write $g_t = \nabla f(\boldsymbol{\theta}_t)$, with $g_{t,i}$ denoting the $i^{\text{th}}$ element of the vector $g_t$. We further write $g_{1:t,i} \in \mathbb{R}^t$ for the $i^{\text{th}}$ dimension of gradients up to iteration $t$.

We begin with the following lemma,

**Lemma 1.** *We write $\mathbf{v}_{t,j}^i$ to indicate the $j^{th}$ element of the $i^{th}$ velocity at time $t$. Assume $g_t$ is bounded, then the following holds for all $j$,*

$$\sum_{t=1}^{T} \sum_{i=1}^{K} \frac{\mathbf{v}_{t,j}^{(i)^2}}{\sqrt{t}} \leq ||g_{1:T,j}||_4^2 \sqrt{1 + \log(T)} \sum_{i=1}^{K} \frac{1}{(1 - \beta^{(i)})^2}$$

**Proof**    We begin by expanding the last term in the sum using the update equations,

$$\sum_{t=1}^{T}\sum_{i=1}^{K}\frac{\mathbf{v}_{t,j}^{(i)2}}{\sqrt{t}} = \sum_{t=1}^{T-1}\sum_{i=1}^{K}\frac{\mathbf{v}_{t,j}^{(i)2}}{\sqrt{t}} + \frac{1}{\sqrt{T}}\sum_{i=1}^{K}\left(\sum_{h=1}^{T}(\beta_{h}^{(i)})^{T-h}g_{h,j}\right)^{2}$$

$$\leq \sum_{t=1}^{T-1}\sum_{i=1}^{K}\frac{\mathbf{v}_{t,j}^{(i)2}}{\sqrt{t}} + \frac{1}{\sqrt{T}}\sum_{i=1}^{K}\left(\sum_{h=1}^{T}(\beta^{(i)})^{T-h}\right)\left(\sum_{h=1}^{T}(\beta^{(i)})^{T-h}g_{h,j}^{2}\right)$$

$$\leq \sum_{t=1}^{T-1}\sum_{i=1}^{K}\frac{\mathbf{v}_{t,j}^{(i)2}}{\sqrt{t}} + \frac{1}{\sqrt{T}}\sum_{i=1}^{K}\frac{1}{1-\beta^{(i)}}\left(\sum_{h=1}^{T}(\beta^{(i)})^{T-h}g_{h,j}^{2}\right)$$

The first inequality is obtained via Cauchy-Schwarz and by noting that $\beta_{t}^{(i)} \leq \beta$ for all $t$. The second inequality follows directly from the fact that $\sum_{h=1}^{T}(\beta^{(i)})^{T-h} < 1/(1-\beta^{(i)})$. We can apply this upper bound to each term of the sum over $t$,

$$\sum_{t=1}^{T}\sum_{i=1}^{K}\frac{\mathbf{v}_{t,j}^{(i)2}}{\sqrt{t}} \leq \sum_{t=1}^{T}\sum_{i=1}^{K}\frac{1}{\sqrt{t}(1-\beta^{(i)})}\sum_{h=1}^{t}(\beta^{(i)})^{t-h}g_{h,j}^{2}$$

$$= \sum_{i=1}^{K}\frac{1}{1-\beta^{(i)}}\sum_{t=1}^{T}\frac{1}{\sqrt{t}}\sum_{h=1}^{t}(\beta^{(i)})^{t-h}g_{h,j}^{2}$$

$$= \sum_{i=1}^{K}\frac{1}{1-\beta^{(i)}}\sum_{t=1}^{T}g_{t,j}^{2}\sum_{h=t}^{T}\frac{(\beta^{(i)})^{h-t}}{\sqrt{h}}$$

$$\leq \sum_{i=1}^{K}\frac{1}{1-\beta^{(i)}}\sum_{t=1}^{T}g_{t,j}^{2}\sum_{h=t}^{T}\frac{(\beta^{(i)})^{h-t}}{\sqrt{t}}$$

$$\leq \sum_{i=1}^{K}\frac{1}{(1-\beta^{(i)})^{2}}\sum_{t=1}^{T}g_{t,j}^{2}\frac{1}{\sqrt{t}}$$

$$\leq \sum_{i=1}^{K}\frac{1}{(1-\beta^{(i)})^{2}}||g_{1:T,j}||_{4}^{2}\sqrt{\sum_{t=1}^{T}\frac{1}{t}}$$

$$\leq ||g_{1:T,j}||_{4}^{2}\sqrt{1+\log(T)}\sum_{i=1}^{K}\frac{1}{(1-\beta^{(i)})^{2}}$$

Under equality we swap the order of sums and collect terms under $g_{t}$. The third inequality follows from $\sum_{j=1}^{t}(\beta^{(i)})^{j-t} < 1/(1-\beta)$. The fourth inequality is an application of Cauchy-Schwarz. The final inequality is from the harmonic sum bound: $\sum_{t=1}^{T}1/t \leq 1+\log(T)$. This completes the proof.

**Proof of Theorem 1**    From the update equations we may write,

$$\boldsymbol{\theta}_{t+1} = \boldsymbol{\theta}_{t} + \frac{\gamma_{t}}{K}\sum_{i=1}^{K}\mathbf{v}_{t}^{(i)}$$

$$= \boldsymbol{\theta}_{t} + \frac{\gamma_{t}}{K}\sum_{i=1}^{K}(\beta_{t}^{(i)}\mathbf{v}_{t-1}^{(i)} - g_{t})$$

We now shift focus to only the $j^{\text{th}}$ dimension. We subtract $\boldsymbol{\theta}_{*j}$ from both sides and square,

$$(\boldsymbol{\theta}_{t+1,j} - \boldsymbol{\theta}_j^*)^2 = (\boldsymbol{\theta}_{t,j} - \boldsymbol{\theta}_j^*)^2 + 2\frac{\gamma_t}{K}(\boldsymbol{\theta}_{t,j} - \boldsymbol{\theta}_j^*)\sum_{i=1}^{K}(\beta_t^{(i)}\mathbf{v}_{t-1,j}^{(i)} - g_{t,j}) + \frac{\gamma_t^2}{K^2}(\sum_{i=1}^{K}\mathbf{v}_{t,j}^{(i)})^2$$

We can rearrange this expression and bound as follows,

$$
\begin{aligned}
g_{t,j}(\boldsymbol{\theta}_{t,j} - \boldsymbol{\theta}_j^*) &= \frac{1}{2\gamma_t}\left[(\boldsymbol{\theta}_{t,j} - \boldsymbol{\theta}_j^*)^2 - (\boldsymbol{\theta}_{t+1,j} - \boldsymbol{\theta}_j^*)^2\right] + (\boldsymbol{\theta}_{t,j} - \boldsymbol{\theta}_j^*)\frac{1}{K}\sum_{i=1}^{K}(\beta_t^{(i)}\mathbf{v}_{t-1,j}^{(i)}) + \frac{\gamma_t}{2K^2}(\sum_{i=1}^{K}\mathbf{v}_{t,j}^{(i)})^2 \\
&= \frac{1}{2\gamma_t}\left[(\boldsymbol{\theta}_{t,j} - \boldsymbol{\theta}_j^*)^2 - (\boldsymbol{\theta}_{t+1,j} - \boldsymbol{\theta}_j^*)^2\right] + \frac{1}{K}\sum_{i=1}^{K}(\boldsymbol{\theta}_{t,j} - \boldsymbol{\theta}_j^*)(\beta_t^{(i)}\mathbf{v}_{t-1,j}^{(i)}) + \frac{\gamma_t}{2K^2}(\sum_{i=1}^{K}\mathbf{v}_{t,j}^{(i)})^2 \\
&= \frac{1}{2\gamma_t}\left[(\boldsymbol{\theta}_{t,j} - \boldsymbol{\theta}_j^*)^2 - (\boldsymbol{\theta}_{t+1,j} - \boldsymbol{\theta}_j^*)^2\right] + \frac{1}{K}\sum_{i=1}^{K}\frac{\sqrt{\beta_t^{(i)}}}{\sqrt{\gamma_{t-1}}}(\boldsymbol{\theta}_{t,j} - \boldsymbol{\theta}_j^*)\sqrt{\gamma_{t-1}\beta_t^{(i)}}(\mathbf{v}_{t-1,j}^{(i)}) \\
&\quad + \frac{\gamma_t}{2K^2}(\sum_{i=1}^{K}\mathbf{v}_{t,j}^{(i)})^2 \\
&\le \frac{1}{2\gamma_t}\left[(\boldsymbol{\theta}_{t,j} - \boldsymbol{\theta}_j^*)^2 - (\boldsymbol{\theta}_{t+1,j} - \boldsymbol{\theta}_j^*)^2\right] + \frac{1}{K}\sum_{i=1}^{K}\frac{\beta_t^{(i)}}{2\gamma_{t-1}}(\boldsymbol{\theta}_{t,j} - \boldsymbol{\theta}_j^*)^2 \\
&\quad + \frac{1}{K}\sum_{i=1}^{K}\frac{\gamma_{t-1}}{2}\beta_t^{(i)}(\mathbf{v}_{t-1,j}^{(i)})^2 + \frac{\gamma_t}{2K^2}(\sum_{i=1}^{K}\mathbf{v}_{t,j}^{(i)})^2 \\
&\le \frac{1}{2\gamma_t}\left[(\boldsymbol{\theta}_{t,j} - \boldsymbol{\theta}_j^*)^2 - (\boldsymbol{\theta}_{t+1,j} - \boldsymbol{\theta}_j^*)^2\right] + \frac{1}{K}\sum_{i=1}^{K}\frac{\beta_t^{(i)}}{2\gamma_{t-1}}(\boldsymbol{\theta}_{t,j} - \boldsymbol{\theta}_j^*)^2 \\
&\quad + \frac{1}{K}\sum_{i=1}^{K}\frac{\gamma_{t-1}}{2}\beta_t^{(i)}(\mathbf{v}_{t-1,j}^{(i)})^2 + \frac{\gamma_t}{2K}\sum_{i=1}^{K}(\mathbf{v}_{t,j}^{(i)})^2
\end{aligned}
$$

The first inequality is an application of Young's inequality. For the second inequality we use the sum-of-squares inequality. We now make use of convexity, and take the sum over dimensions and time,

$$\sum_{t=1}^{T} f_t(\boldsymbol{\theta}_t) - f_t(\boldsymbol{\theta}^*) \leq \sum_{t=1}^{T} \sum_{j=1}^{d} g_{t,j}(\boldsymbol{\theta}_{t,j} - \boldsymbol{\theta}_j^*)$$

$$\leq \sum_{t=1}^{T} \sum_{j=1}^{d} \frac{1}{2\gamma_t} \left[ (\boldsymbol{\theta}_{t,j} - \boldsymbol{\theta}_j^*)^2 - (\boldsymbol{\theta}_{t+1,j} - \boldsymbol{\theta}_j^*)^2 \right] + \frac{1}{K} \sum_{i=1}^{K} \frac{\beta_t^{(i)}}{2\gamma_{t-1}} (\boldsymbol{\theta}_{t,j} - \boldsymbol{\theta}_j^*)^2$$

$$+ \frac{1}{K} \sum_{i=1}^{K} \frac{\gamma_{t-1}}{2} \beta_t^{(i)} (\mathbf{v}_{t-1,j}^{(i)})^2 + \frac{\gamma_t}{2K} \sum_{i=1}^{K} (\mathbf{v}_{t,j}^{(i)})^2$$

$$\leq \sum_{j=1}^{d} \frac{1}{2\gamma_1} (\boldsymbol{\theta}_{1,j} - \boldsymbol{\theta}_j^*)^2 + \frac{1}{2} \sum_{j=1}^{d} \sum_{t=1}^{T} (\boldsymbol{\theta}_{t,j} - \boldsymbol{\theta}_j^*)^2 (\frac{1}{\gamma_t} - \frac{1}{\gamma_{t-1}})$$

$$+ \frac{\gamma \sqrt{1 + \log(T)}}{2K} \sum_{j=1}^{d} ||g_{1:T,j}||_4^2 \sum_{i=1}^{K} \frac{1 + \beta^{(i)}}{(1 - \beta^{(i)})^2}$$

$$+ \frac{1}{K} \sum_{i=1}^{K} \sum_{j=1}^{d} \sum_{t=1}^{T} \frac{\beta_t^{(i)}}{2\gamma_{t-1}} (\boldsymbol{\theta}_{t,j} - \boldsymbol{\theta}_j^*)^2$$

We now make use of the bounding assumptions, $||\boldsymbol{\theta}_m - \boldsymbol{\theta}_n||_2 \leq D$ and $||\boldsymbol{\theta}_m - \boldsymbol{\theta}_n||_\infty \leq D_\infty$,

$$R(T) \leq \frac{D_\infty^2 \sqrt{T}}{\gamma} + \frac{\gamma \sqrt{1 + \log(T)}}{2K} \sum_{j=1}^{d} ||g_{1:T,j}||_4^2 \sum_{i=1}^{K} \frac{1 + \beta^{(i)}}{(1 - \beta^{(i)})^2} + \frac{D^2}{2\gamma} \frac{1}{K} \sum_{i=1}^{K} \sum_{t=1}^{T} \beta^{(i)} \lambda^{t-1} \sqrt{t}$$

The first two terms are collapsed using a telescoping sum. Using $\sum_t \lambda^{t-1} \sqrt{t} \leq 1/(1 - \lambda)^2$, we achieve the following bound,

$$R(T) \leq \frac{D_\infty^2 \sqrt{T}}{\gamma} + \frac{\gamma \sqrt{1 + \log(T)}}{2K} \sum_{j=1}^{d} ||g_{1:T,j}||_4^2 \sum_{i=1}^{K} \frac{1 + \beta^{(i)}}{(1 - \beta^{(i)})^2} + \frac{D^2}{2K\gamma(1 - \lambda)^2} \sum_{i=1}^{K} \beta^{(i)}$$

## C.1 Open Questions on Convergence

While studying the convergence properties of AggMo we made several interesting observations which presented theoretical challenges. We present some of these observations here to shed light on key differences between AggMo and existing momentum methods. We hope that these will provoke further study.

**Further reduction of** $B$    In Appendix B we derived the matrix $B$ in order to get bounds on the convergence. We can further reduce $B$ to block diagonal form, where the $j^{th}$ block takes the form,

$$B_j = \begin{bmatrix} \beta^{(1)} & 0 & \cdots & 0 & -\lambda_j \\ 0 & \beta^{(2)} & \ddots & \vdots & \vdots \\ \vdots & \ddots & \ddots & 0 & -\lambda_j \\ 0 & \cdots & 0 & \beta^{(K)} & -\lambda_j \\ \frac{\gamma \beta^{(1)}}{K} & \frac{\gamma \beta^{(2)}}{K} & \cdots & \frac{\gamma \beta^{(K)}}{K} & (1 - \gamma \lambda_j) \end{bmatrix}$$

From this relatively simple form we may be able to derive a closed-form solution for the eigenvalues which would allow us to reason theoretically about the quadratic convergence properties of AggMo. An easier goal would be finding suitable conditions under which the eigenvalues are complex and the system is under-damped.

**Finite Difference Equation**   In this section we demonstrate that the dynamics of AggMo can be written as a $(K+1)$-th order finite difference equation. While most momentum methods can be viewed as the discretization of second order ODEs (Wilson et al., 2016) it seems that AggMo does not fall into this class of algorithms. As a consequence, it becomes difficult to apply existing convergence proof techniques to AggMo.

For simplicity, we assume a fixed learning rate $\gamma$ for all time steps. We will first tackle the special case $K = 2$. From the AggMo update rule, we have

$$
\begin{bmatrix} \mathbf{v}_{t+1}^{(1)} \\ \mathbf{v}_{t+1}^{(2)} \\ \mathbf{v}_t^{(1)} \\ \mathbf{v}_t^{(2)} \\ \mathbf{v}_{t-1}^{(1)} \\ \mathbf{v}_{t-1}^{(2)} \end{bmatrix} = \begin{bmatrix} 0 & 0 & \beta_1 & 0 & 0 & 0 \\ 0 & 0 & 0 & \beta_2 & 0 & 0 \\ 0 & 0 & 0 & 0 & \beta_1 & 0 \\ 0 & 0 & 0 & 0 & 0 & \beta_2 \\ 0 & 0 & \frac{\gamma}{K} & \frac{\gamma}{K} & 1 & 0 \\ 0 & 0 & 0 & 0 & \frac{\gamma}{K} & 1+\frac{\gamma}{K} \end{bmatrix} \begin{bmatrix} \mathbf{v}_{t+1}^{(1)} \\ \mathbf{v}_{t+1}^{(2)} \\ \mathbf{v}_t^{(1)} \\ \mathbf{v}_t^{(2)} \\ \mathbf{v}_{t-1}^{(1)} \\ \mathbf{v}_{t-1}^{(2)} \end{bmatrix} - \begin{bmatrix} \nabla_\theta f(\theta_t) \\ \nabla_\theta f(\theta_t) \\ \nabla_\theta f(\theta_{t-1}) \\ \nabla_\theta f(\theta_{t-1}) \\ \theta_t - \theta_{t-1} \\ \theta_{t-1} - \theta_{t-2} \end{bmatrix} \tag{9}
$$

Denoting the matrices as symbols correspondingly, it becomes $\mathbf{v} = \mathbf{B}\mathbf{v} - \mathbf{g}$, therefore

$$
\mathbf{v} = -(\mathbf{I} - \mathbf{B})^{-1}\mathbf{g} \tag{10}
$$

Denote $\delta_t = \theta_t - \theta_\star$, then $\theta_t - \theta_{t-1} = \delta_t - \delta_{t-1}$. Note that $\theta_{t+1} = \theta_t + \frac{\gamma_{t+1}}{2}(\mathbf{v}_{t+1}^{(1)} + \mathbf{v}_{t+1}^{(2)})$, plugging Eq 10 into it, we have

$$
\delta_{t+1} = \delta_t - \frac{\gamma}{2}[1, 1, 0, 0, \cdots]^\top (\mathbf{I} - \mathbf{B})^{-1}\mathbf{g} \tag{11}
$$

Which reduces to the following finite difference equation,

$$
\delta_{t+1} = (1 + \beta_1 + \beta_2)\delta_t + (\beta_1 + \beta_2 + \beta_1\beta_2)\delta_{t-1} - \beta_1\beta_2\delta_{t-2} + \frac{\gamma}{2}(2\nabla_\theta f(\theta_t) - (\beta_1 + \beta_2)\nabla_\theta f(\theta_{t-1})) \tag{12}
$$

For $K \geq 2$, we only need to change Eq 9 accordingly, follow the remaining derivations, and recover a $(K+1)$-th order difference equation. We could also derive the same result using sequence elimination, made simpler with some sensible variable substitutions.

This result is of considerable importance. Existing momentum methods can generally be rewritten as a second order difference equation (Section 2 in O'Donoghue & Candes (2015)) which then induce a second order ODE (Su et al., 2014; Wibisono & Wilson, 2015). The momentum optimization procedure can then be thought of as a discretization of a Hamiltonian flow. On the other hand, AggMo does not obviously lend itself to the analytical tools developed in this setting - it is not obvious whether the form in AggMo is indeed a discretization of a Hamiltonian flow.

# D   Experiments

All of our experiments are conducted using the pytorch library Paszke et al. (2017). In each experiment we make use of early stopping to determine the run with the best validation performance.

## D.1   Autoencoders

For the autoencoders we train fully connected networks with encoders using the following architecture: 784-1000-500-250-30. The decoder reverses this architecture. We use relu activations throughout the network. We train for a total of 1000 epochs using a multiplicative learning rate decay of 0.1 at 200, 400, and 800 epochs. We train using batch sizes of 200.

For these experiments the training set consists of 90% of the training data with the remaining 10% being used for validation.

For each optimizer we searched over the following range of learning rates: { 0.1, 0.05, 0.01, 0.005, 0.001, 0.0005, 0.0001, 0.00005, 0.00001}.

## D.2 Classification

For each of the classification tasks we train for a total of 400 epochs using batchsizes of 128. We make use of a multiplicative learning rate decay of 0.1 at 150 and 250 epochs. For each of these experiments we use 80% of the training data for training and use the remaining 20% as validation.

In these experiments we searched over the following learning rates for all optimizers: { 0.1, 0.05, 0.01, 0.005, 0.001, 0.0005, 0.0001 }. We searched over the same damping coefficients as in the autoencoder experiments. Each model was trained for a total of 500 epochs.

When training without batch normalization we explored a smaller range of learning rates for both CM and AggMo: { 0.1, 0.05, 0.01, 0.005 }.

**CNN-5** The CNN-5 model uses relu activations throughout and 2x2 max pooling with stride 2. The first convolutional layer uses an 11x11 kernel with a stride of 4. This is followed by a max pooling layer. There is then a 5x5 convolutional kernel followed by max pooling. The network then uses three 3x3 convolutional layers and a final max pooling layer before feeding into a fully connected output layer. We do not use any regularization when training this model.

**ResNet-32** We use the ResNet-32 architecture on both CIFAR-10 and CIFAR-100. We make use of a weight decay of 0.0005 and use batch normalization (Ioffe & Szegedy, 2015). We introduce data augmentation by using random crops with a padding of 4 and use random horizontal flips with probability 0.5.

## D.3 LSTM Language Modelling

We train LSTMs with 3-layers containing 1150 hidden units per layer, and a 400 embedding size. Within the network we use dropout on the layers with probability 0.4. The hidden layers use dropout with probability 0.3 and the input embedding layers use dropout with probability 0.65 while the embedding layer itself uses dropout with probability 0.1. We also apply the weight drop method proposed in Merity et al. (2017) with probability 0.5. L2 regularization is applied on the RNN activations with a scaling of 2.0, we also use temporal activation regularization (slowness regularization) with scaling 1.0. Finally, all weights receive a weight decay of 1.2e-6.

We train the model using variable sequence lengths and batch sizes of 80. We measure the validation loss during training and decrease the learning rate if the validation loss has not decreased for 15 epochs. We found that a learning rate decay of 0.5 worked best for all optimizers except for SGD which achieved best performance with a fixed learning rate.

For SGD, CM, AggMo and Nesterov we searched over learning rates in the range {50, 30, 10, 5, 2.5, 1, 0.1, 0.01}. We found that Adam required much smaller learning rates in this setting and so searched over values in the range {0.1, 0.05, 0.01, 0.005, 0.001, 0.0005, 0.0001}. We searched over the damping coefficients as in the previous experiments. Each model was trained for 750 epochs, as in Merity et al. (2017).

# E Additional Results

In this section we display some of the experimental results which we are unable to fit in the main paper.

## E.1 Toy Problem

To better understand how AggMo is able to help in non-convex settings we explore its effectiveness on a simple non-convex toy problem. The function we aim to optimize is defined as follows,

$$
\begin{aligned}
f(x, y) = {}&\log(e^x + e^{-x}) + \\
& b \log\left(e^{e^x(y - \sin(ax))} + e^{-e^x(y - \sin(ax))}\right)
\end{aligned}
\tag{13}
$$

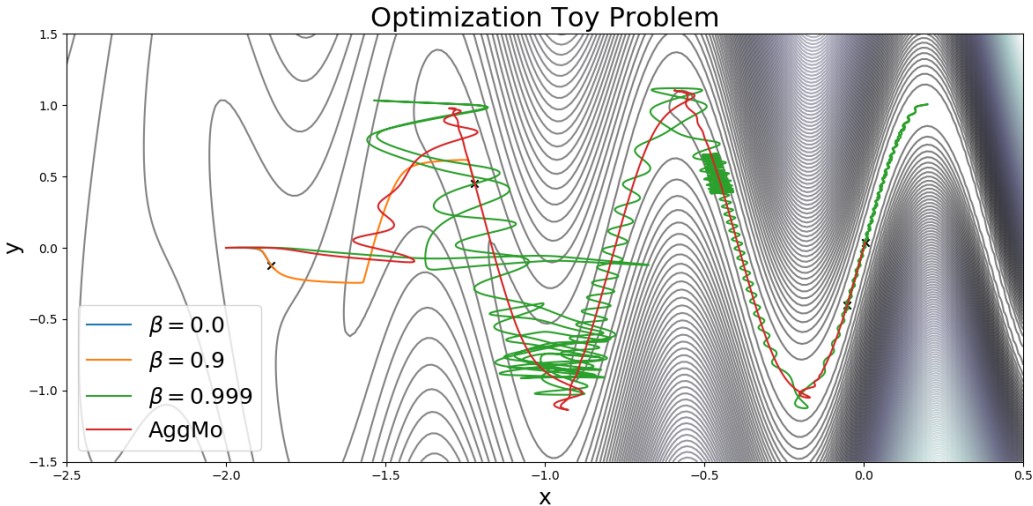

Figure 11: Comparison of classical momentum and aggregated momentum on toy problem (13) with $a = 8, b = 10$. In each case the optimizer is initialized at $(x, y) = (-2, 0)$

| Optimizer | Train Optimal | Validation Optimal | |
| --- | --- | --- | --- |
| | Train Loss | Val. Loss | Test Loss |
| **CM** $\beta = 0.9$ | 2.07 | 4.95 | 4.98 |
| **Nesterov** $\beta = 0.9$ | 1.94 | 4.63 | 4.62 |
| **AggMo** (Default) | **1.60** | **3.14** | **3.04** |

Table 4: **MNIST Autoencoder with default settings** We display the training MSE for the initial learning rate that achieved the best training loss. The validation and test errors are displayed for the initial learning rate that achieved the best validation MSE.

where $a$ and $b$ are constants which may be varied. We choose this function because it features flat regions and a series of non-convex funnels with varied curvature. The optimizer must traverse the flat regions quickly whilst remaining stable within the funnels. This function has an optimal value at $(x, y) = (0, 0)$.

Figure 11 compares the performance of classical momentum and aggregated momentum when optimizing Equation 13 with $a = 8, b = 10$. We see that GD with $\beta = 0$ and $\beta = 0.9$ are unable to leave the flat region around $x < -1$. For GD with $\beta = 0.999$ the optimizer enters the funnels but frequently becomes unstable with oscillations and finally overshoots the optimum. Compared to GD, AggMo is able to quickly traverse both the flat region and the funnels while remaining stable. AggMo also successfully slows down quickly once reaching the optimum.

## E.2 Comparison at default damping settings

In this section we present results using the default damping coefficient settings for the autoencoder and LSTM experiments.

The default settings for CM, Nesterov, and AggMo are compared in Table 4. The default settings of AggMo outperform both CM and Nesterov significantly. Moreover, while the AggMo default settings perform similarly to the best results in Table 1 there is a large gap for the CM and Nesterov defaults. This suggests that for this task AggMo is less sensitive to hyperparameter tuning than the other methods.

For the LSTM experiments we found that all methods worked best with their default damping coefficients except for Nesterov momentum which used $\beta = 0.99$. For Nesterov momentum with $\beta = 0.9$ the validation perplexity was 63.67 and the test perplexity was 61.45. AggMo with default settings achieved better training, validation and test perplexity than both the CM and Nesterov defaults.

# F    Beta-Averaged Momentum

In this section we present a continuous analog of AggMo which provides additional insight into its effectiveness.

The AggMo update rule features the average of several velocities with some chosen damping coefficients, $\boldsymbol{\beta}$. A natural extension to this formulation instead considers a mapping from beta values to velocities with the space of velocities being integrated over instead of summed. Explicitly, we write this update rule as,

$$
\begin{aligned}
\mathbf{v}_t &= b\mathbf{v}_{t-1} - \nabla_\theta f(\boldsymbol{\theta}_{t-1}) \\
\boldsymbol{\theta}_t &= \boldsymbol{\theta}_{t-1} + \gamma \int_0^1 \mathbf{v}_t \pi(b) db
\end{aligned}
\tag{14}
$$

Where $\pi(b)$ is a probability density defined on $[0, 1]$. We can link this back to aggregated momentum in the following way. If we sampled $b^{(i)}$ under the density $\pi$ for $i = 1 : M$ then the procedure described by Equation 3 is approximating Equation 14 via Monte Carlo Integration.

Although this seems like a reasonable idea, it is not obvious whether we can compute this integral in closed form. We can understand this update rule by expanding $\mathbf{v}_t$ recursively,

$$
\begin{aligned}
\mathbf{v}_t &= b\mathbf{v}_{t-1} - \nabla_\theta f(\boldsymbol{\theta}_{t-1}) \\
&= b(b\mathbf{v}_{t-2} - \nabla_\theta f(\boldsymbol{\theta}_{t-2})) - \nabla_\theta f(\boldsymbol{\theta}_{t-1}) \\
&= b^t \mathbf{v}_0 - \sum_{i=1}^{t} b^{i-1} \nabla_\theta f(\boldsymbol{\theta}_{t-i}) \\
&= -\sum_{i=1}^{t} b^{i-1} \nabla_\theta f(\boldsymbol{\theta}_{t-i}) = -\sum_{i=0}^{t-1} b^{t-i-1} \nabla_\theta f(\boldsymbol{\theta}_i)
\end{aligned}
\tag{15}
$$

Thus we can write the update rule for $\mathbf{x}_t$ as,

$$
\boldsymbol{\theta}_t = \boldsymbol{\theta}_{t-1} - \gamma \sum_{i=1}^{t} \nabla_\theta f(\boldsymbol{\theta}_{t-i}) \int_0^1 b^{i-1} \pi(b) db
\tag{16}
$$

Thus to compute the update rule we must compute the raw moments of $b$. Fortunately, for the special case where $\pi$ is the density function of a Beta distribution then we have closed form solutions for the raw moments of $b \sim Beta(\alpha, \beta)$ (note that $\beta$ here is **not** referring to a damping coefficient) then these raw moments have a closed form:

$$
\mathbb{E}[b^k] = \prod_{r=0}^{k-1} \frac{\alpha + r}{\alpha + \beta + r}
\tag{17}
$$

This provides a closed form solution to compute $\boldsymbol{\theta}_t$ given $\boldsymbol{\theta}_{t-1}$ and the history of all previous gradients. We refer to this update scheme as *Beta-Averaged Momentum*. Unfortunately, each update requires the history of all previous gradients to be computed. We may find some reasonable approximation to the update rule. For example, we could keep only the $T$ most recently computed gradients.

Figure 12 shows the optimization of 1D quadratics using Beta-Averaged Momentum. The trajectories are similar to those achieved using the original AggMo formulation.

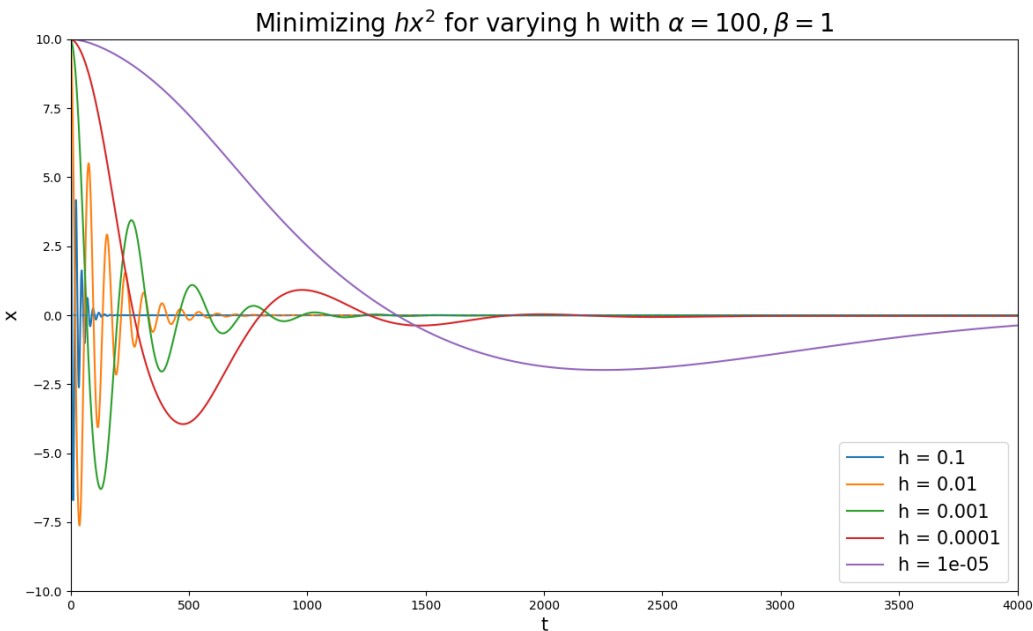

Figure 12: Beta-Averaged GD with a Beta prior on momentum ($\alpha = 100, \beta = 1$).

