# OpenReview forum: "Aggregated Momentum: Stability Through Passive Damping"
_ICLR.cc/2019/Conference_

### Official Review · AnonReviewer2 · 2018-10-29
**Cool Idea but concerns about experiments**

**Rating:** 5
**Confidence:** 4

**Review:**

The authors combined several update steps together to achieve aggregated momentum. They showed that  it is more stable than the other momentum methods. Also, in Auto-encoder and image classification, AggMo outperforms than the other methods.

Pros:
(+) Theoretical result is shown on the quadratic problem.

(+) Extensive numerical experiments are shown to illustrate the stability of AggMo.

Cons:
(+) The results are not convincing. For example, it said in the default setting (CM \beta=0.9), ResNet34 on CIFAR-10 has accuracy 90.22\%. However, it should be around 93\%.

(+)  This method is similar to multi-step gradient methods.



Comments:
(+) This is no “introduction” in the paper.

(+) There should be “,” after mathematical equations.

---

> ### Author Response · Authors · 2018-11-17
> **Thank you for pointing out this error. Revised paper has been corrected.**
>
> Thank you for your feedback!
>
> #1 Empirical Results
>
> We thank you in particular for pointing out the issues with the ResNet-34 results. After investigating these further we were able to confirm a bug in our implementation, causing batch norm statistics to become fixed after a single epoch of training. We reran the experiments with this bug fixed and have updated the paper to reflect this. Due to computational limitations, we replaced the ResNet-34 architecture with ResNet-32.
>
> To summarize, CM is now able to achieve similar final accuracy to AggMo but AggMo retains faster convergence throughout training. We are confident that the numbers reported are correct and in fact we report almost a full percentage higher accuracy than the original ResNet paper on CIFAR-10 with CM (they reported 92.49%).
>
> The bug also prompted us to rerun these experiments with batch norm completely disabled. We found that AggMo is able to remain stable over a large range of learning rates even without batch normalization. Without batch normalization, AggMo achieves a final test accuracy on CIFAR-100 of 69.32%, more than 2% higher than the best value achieved by CM.
>
> We apologize for the issue with the original results but want to emphasize that this did not affect any other empirical results. We would also like to point out that AggMo performed consistently well across all tasks and is able to do so over a wide range of hyperparameters settings. We believe that the results as a whole clearly show that AggMo is a powerful optimizer that often outperforms well established methods, even on tasks that have been developed with the original methods in mind (e.g. the model hyperparameters we used are chosen to work well with CM). Less formally, in other settings we have applied AggMo to a huge variety of problems and have yet to find a failure case. We would be happy to address any other specific concerns you have about the empirical results.
>
> #2 Similarity to multi-step gradient methods
>
> Could you please clarify what you mean by “multi-step gradient methods”? After looking ourselves, this seemed to be a less common term for multi-state algorithms such as CM or Nesterov. We see this as a pro and not a con of our algorithm. For example, much of the intuition for tuning hyperparameters for Nesterov/CM carries over to AggMo. Reviewer 3 raised a similar concern on the incremental nature of AggMo. We have responded to them but repeat that argument here. We have intentionally presented AggMo in a simple form which makes it easily relatable to existing algorithms. However, AggMo realizes complex dynamics which in general escape many theoretical frameworks. In Appendix C.2 we show that AggMo can be written as a (K+1)th order finite difference equation. We agree that AggMo is a simple extension of classical momentum but emphasize that most successful optimization algorithms are --- the challenge is in finding simple variants which perform well.
>
> #3 Minor comments
>
> We have punctuated the equations and added in the “Introduction” section heading as requested.
>
> Thank you again for your review. We hope that our above comments and new empirical results address your concerns. We are happy to discuss any other concerns raised in your response.

---

> > ### Comment · AnonReviewer2 · 2018-11-27
> > **Thanks for fixing the Numerical Results**
> >
> > First, thanks a lot to fix the numerical results and the typos.
> >
> > For the second part, please take a look at arxiv: 1211.2132, 1606.02118. Could you please point out the main differences between AggMo and those methods?

---

> > > ### Author Response · Authors · 2018-11-27
> > > **Comparing to these papers**
> > >
> > > Thank you for your response and for the interesting references.
> > >
> > > For the first paper (“Accelerated Gradient Methods for Networked Optimization“) we do not consider this work to share significant overlap with AggMo. This work studies the networked optimization problem where there are several convex optimization objectives which must be solved subject to some (linear) constraints. The authors adapt classical momentum to this setting and prove convergence properties. The primary contribution seems to be handling this constrained setting. To quote the authors: “In the absence of constraints, (1) is trivial to solve [...]”.
> > >
> > > We believe that the second paper is more closely related (“A Multi-step Inertial Forward–Backward Splitting Method for Non-convex Optimization”) but we still identify some critical differences. Algorithm 1 (MiFB) from this work introduces longer time dependence of the acceleration terms which we show AggMo also achieves in Appendix C.2. However, MiFB is parameterized by many more hyperparameters than AggMo (more than twice as many). The form of the loss is also decomposed into two terms, R and F, and MiFB uses two sequences (y_a and y_b), the second of which is used only for the gradient evaluation of F. Importantly, MiFB uses a proximity operator to push the iterates into a feasible set based on R --- something which we do not consider in AggMo. Finally, each iterate update uses only the current gradient update explicitly, while AggMo also uses a history of the gradients for each update (see Appendix C.2). To summarize, MiFB is stated very generally and due to the large number of parameters we believe that with some modifications it can probably recover AggMo-esque updates. However, if true, we expect this would hold for the vast majority of first order optimization algorithms and therefore do not consider this comparison to be especially informative.
> > >
> > > We suspect that you are much more an expert on this setting than us and would be happy to receive any corrections on our interpretation of these papers.
> > >
> > > Thank you for acknowledging the corrected numerical results. If you have no other concerns, we kindly ask that you consider raising your review score. Otherwise, we would be happy to answer any other questions you may have.

---

### Official Review · AnonReviewer3 · 2018-11-02
**Interesting but incremental**

**Rating:** 6
**Confidence:** 3

**Review:**

This paper proposed an aggregated momentum methods for gradient based optimization. The basic idea is instead of using a single velocity vector, multiple velocity vectors with different damping factors are used in order to improve the stability.

In term of novelty, the proposed method seems quite incremental. Using multiple velocity vectors seems interesting but not surprising, There is no theoretical guideline how to determine the number of velocity vectors and how to choose the damping factors.

I would also suggest that authors should put some main theoretical results like the convergence analysis to the main paper instead of the appendix.

In terms of the clarity, I think the paper is well written and the experiments are sufficient and convincing.

One minor question is: what is \lambda in Fig. 1?

---

> ### Author Response · Authors · 2018-11-17
> **Thank you!**
>
> Thank you for your feedback!
>
> We appreciate your honest comments on novelty, but respectfully disagree. We have intentionally presented AggMo in a simple form which makes it easily relatable to existing algorithms. However, AggMo realizes complex dynamics which in general escape many theoretical frameworks. In Appendix C.2 we show that AggMo can be written as a (K+1)-th order finite difference equation. We agree that AggMo is a simple extension of classical momentum but emphasize that most successful optimization algorithms are --- the challenge is in finding simple variants which perform well.
>
> While we do not present theoretical guidelines for choosing damping factors, we conduct an extensive empirical study and suggest a range of choices which we found worked well consistently. Choosing optimization hyperparameters is a huge challenge in deep learning, even with well known, extensively explored optimizers. For example, while practitioners typically use a damping coefficient of 0.9 with CM we found that we could achieve better performance with larger coefficients (and carefully tuned learning rates). AggMo is able to use larger damping coefficients without such extensive tuning. Furthermore, AggMo is able to recover CM, NAG, and other algorithms and can make direct use of theoretical convergence results for these.
>
> Per your suggestion, we have moved the online convex programming theoretical results into the main text.
>
>
>
> > What is \lambda in Fig. 1?
>
> This indicates the eigenvalue associated with the plotted eigendirection trajectory. This is detailed in the main text but we have added a note to the caption.

---

> ### Comment · Area_Chair1 · 2018-11-27
> **Comment on "surprising"**
>
> "Using multiple velocity vectors seems interesting but not surprising": I am not aware of works using a similar technique, despite momentum dating back to 1964. As a result, I am not sure I understand your comment. Could you please explain?

---

### Official Review · AnonReviewer1 · 2018-11-03
**Nice algorithm that is simple yet effective and has good intuition**

**Rating:** 7
**Confidence:** 3

**Review:**

The paper introduces a variant of momentum that aggregates several velocities with different dampening coefficients. The proposed optimization algorithm can significantly decrease oscillation thus one can use much larger dampening coefficient to achieve faster convergence.

The paper does a good job of motivating the algorithm, both in terms of intuitive reasoning from physics, and some demonstrative examples. The paper also has nice analysis in the simple quadratic case where it tries to make equivalence to the Nesterov's accelerated gradients.

The experiments are also thorough and convincing since it contains various network architectures and different datasets. From the comparison, it seems AggMo consistently achieves the best or comparable performance even in test error/accuracies.

The connection to Nesterov's accelerated gradient and extragradient methods can be discussed in more details. It'll be nice to put the theoretical results in the main text as well.

---

> ### Author Response · Authors · 2018-11-17
> **Thank you! Feedback integrated into paper.**
>
> Thank you for your review!
>
> We are glad that you found the experimental evidence convincing! For transparency, Reviewer 2 pointed out an issue with our reported numbers for the ResNet34 experiments. We have corrected these and provide a detailed response to Reviewer 2 directly. To summarize, AggMo still achieves the fastest overall convergence by a substantial margin but the baseline methods now match in final validation performance. All other empirical results are unaffected.
>
> We have added some further discussion of the connection between AggMo and Nesterov momentum. If there is anything else that you would like to see we would be pleased to be told explicitly. We have also moved the online convex programming theoretical result in the main text. Thank you for the suggestion!

---

### Author Response · Authors · 2018-11-17
**Revision and comments uploaded**

Thank you to each reviewer for taking the time to evaluate our work!

We have revised our submission and provided detailed responses to each of you. We hope that these updates address your concerns and look forward to your responses.

---

### Meta-Review · Area_Chair1 · 2018-12-13
**A new take on momentum which deserves a longer assessment of related work**

**Confidence:** 4
**Recommendation:** Accept (Poster)

**Metareview:**

Dear authors,

Reviewers liked the idea of your new optimizer and found the experiments convincing. However, they also would have liked to get better insights on the place of AggMo in the existing optimization literature. Given that the related work section is quite small, I encourage you to expand it based on the works mentioned in the reviews.